Identification of potential gene signatures associated with osteosarcoma by integrated bioinformatics analysis

Jia Yutao 1
Liu Yang 1
Han Zhihua hanzhihua1996@163.com 2
Tian Rong 1
1 Department of Spine Surgery, Tianjin Union Medical Center , Tianjin , China
2 Department of Anesthesiology, Tianjin Union Medical Center , Tianjin , China
Driscoll Timothy
Electronic publication date: 2021 May 27
Publication date: 2021
Volume: 9
Electronic Location ID: e11496
Received 2020 Jul 1; Accepted 2021 Apr 30
Copyright: ©2021 Jia et al.
Copyright year: 2021
Copyright holder: Jia et al.
License: This is an open access article distributed under the terms of the Creative Commons Attribution License, which permits unrestricted use, distribution, reproduction and adaptation in any medium and for any purpose provided that it is properly attributed. For attribution, the original author(s), title, publication source (PeerJ) and either DOI or URL of the article must be cited.
License URL: https://creativecommons.org/licenses/by/4.0/

Keywords: Osteosarcoma, Diagnosis, Genes, Activating transcription factor, Bioinformatic

Funding: The Tianjin Union Medical Center Research Project 2019JZPY06, 2019JZPY02 This work was supported by the Tianjin Union Medical Center Research Project2019JZPY06, 2019JZPY02. The funders had no role in study design, data collection and analysis, decision to publish, or preparation of the manuscript.

==============================
Background

Osteosarcoma (OS) is the most primary malignant bone cancer in children and adolescents with a high mortality rate. This work aims to screen novel potential gene signatures associated with OS by integrated microarray analysis of the Gene Expression Omnibus (GEO) database.

Material and Methods

The OS microarray datasets were searched and downloaded from GEO database to identify differentially expressed genes (DEGs) between OS and normal samples. Afterwards, the functional enrichment analysis, protein–protein interaction (PPI) network analysis and transcription factor (TF)-target gene regulatory network were applied to uncover the biological function of DEGs. Finally, two published OS datasets (GSE39262 and GSE126209) were obtained from GEO database for evaluating the expression level and diagnostic values of key genes.

Results

 In total 1,059 DEGs (569 up-regulated DEGs and 490 down-regulated DEGs) between OS and normal samples were screened. Functional analysis showed that these DEGs were markedly enriched in 214 GO terms and 54 KEGG pathways such as pathways in cancer. Five genes (CAMP, METTL7A, TCN1, LTF and CXCL12) acted as hub genes in PPI network. Besides, METTL7A, CYP4F3, TCN1, LTF and NETO2 were key genes in TF-gene network. Moreover, Pax-6 regulated four key genes (TCN1, CYP4F3, NETO2 and CXCL12). The expression levels of four genes (METTL7A, TCN1, CXCL12 and NETO2) in GSE39262 set were consistent with our integration analysis. The expression levels of two genes (CXCL12 and NETO2) in GSE126209 set were consistent with our integration analysis. ROC analysis of GSE39262 set revealed that CYP4F3, CXCL12, METTL7A, TCN1 and NETO2 had good diagnostic values for OS patients. ROC analysis of GSE126209 set revealed that CXCL12, METTL7A, TCN1 and NETO2 had good diagnostic values for OS patients.

Introduction

Osteosarcoma (OS) is a type of primary malignant bone cancer that causes public health concern, especially in children and adolescents (Isakoff, Meltzer & Gorlick, 2015; Lindsey, Markel & Kleinerman, 2017). Several treatment strategies of OS such as surgical resection, traditional adjuvant chemotherapy and radiotherapy have been favored by clinical oncologists in the past few decades (Nagarajan et al., 2011). Accordingly, the 5-year survival rate has been raised to approximately 70% (Bielack et al., 2002). However, it is estimated that 80% of OS patients may suffer from the micro-metastasis, which cannot be detected at early diagnosis (Messerschmitt et al., 2009). Although multiple methods for the diagnosis and treatment of OS have been developed, new methods for the prevention and treatment of OS are still needed. The pathogenesis of OS progression remains not fully understood. Therefore, the identification of effective diagnostic makers and exploring underlying molecular etiology of OS is a pressing need.

The emergence of high-throughput sequencing technology has become an effective way to illuminate the pathogenic genes in a variety of human diseases, which help to explore pathogenesis and develop biomarkers. Many research groups have screened multiple biomarkers associated with OS by analyzing gene expression data. For example, Sun et al. evaluated the difference of genes in the expression level between OS metastasis and OS non-metastasis and discovered that TGFB1, LFT3, KDM1A, and KRAS may participate in the occurrence of OS. Xiong et al. (2015) found that CCT3, COPS3 and WWP1 were involved in the OS development by integrating gene expression and genomic aberration data. Liu, Zhao & Chen (2017) constructed a co-expression network based on a Gene Expression Omnibus (GEO) dataset and identified many potential biomarkers such as CTLA4 and PBF for diagnosis and treatment of OS. However, the molecular mechanisms of OS initiation and development have not been fully explored.

In this study, we retrieved GEO database and obtained four OS datasets. Subsequently, the differentially expressed genes (DEGs) between OS samples and normal samples were obtained and subjected to functional analysis. A protein–protein interaction (PPI) network was constructed followed by the establishment of transcription factor (TF)-target regulatory network. Following this, we downloaded two published GEO datasets of OS as the validation set for assessing the expression levels of key candidate genes. Finally, the receiver operating characteristic (ROC) analysis was conducted to evaluate the diagnostic values of key candidate genes. This study will discover novel gene signatures associated with OS, providing new trains of thought for the diagnosis and treatment of OS.

Materials and Methods

Data acquisition

The datasets were retrieved from the National Center for Biotechnology Information-GEO (http://www.ncbi.nlm.nih.gov/geo/) repository using the key terms of ‘osteosarcoma’ AND ‘Homo sapiens’[porgn]. All selected datasets in this study should meet the following criteria: (i) the datasets contained genome-wide expression data of tumor tissues and normal control tissues of OS patients; and (ii) all data were standardized or raw data. As shown in Table S1, a total of five datasets were obtained. Notably, the GSE9508 dataset contained over 50% missing data and was subsequently removed from the following analysis. Eventually, four datasets (GSE12865, GSE19276, GSE87624 and GSE99671) were included in this study, which included 118 OS tissues and 28 normal bone tissues. The GSE12865 series (GPL6244 platform) included a total of 14 samples (12 OS and two normal control tissues). The platform for GSE19276 was GPL6848 including 44 OS and five normal control bone tissue. The platform for GSE87624, consisting of 44 OS and three normal control bone tissue samples, was GPL11154. GSE99671 was in GPL20148 platform, which contained 18 OS and 18 normal control bone tissue samples. The platform and series matrix files were downloaded. The dataset information was listed in Table S1. The impact of different platforms on the results, we normalized the data through the log function, and centralized and standardized the scale function to eliminate the impact of the dimension on the data structure.

Data pre-processing and DEGs identification

The standardized data from included datasets were firstly processed as follows: (i) the probes that mapped to several genes were deleted; and (ii) if the gene was matched by multiple probes, the probe with the greatest gene expression value would be retained. Overall, there were overlapping 14981 genes among four datasets. Subsequently, MetaMA (https://cran.r-project.org/web/packages/metaMA/), an R package, was used to combine data from four GEO datasets. Individual p-values were obtained using Limma R package. The inverse normal method was used to combine P values in meta-analysis (Marot, Mayer & Jaffrézic, 2009). We carried out the multiple comparison correction by Benjamini & Hochberg approach to acquire false discovery rate (FDR). Herein, the DEGs between OS tissues and normal controls were defined according to the cutoff of false discovery rate (FDR) <  0.05 and those DEGs with different directionality were removed from this study. Finally, the hierarchical clustering analysis of top 100 DEGs was also carried out by pheatmap package in R software.

Functional enrichment analyses

To systematically explore the underlying biological functions of identified DEGs, the Metascape (http://metascape.org/gp/index.html), an online tool that integrates multiple data resources such as Gene Ontology (GO), Kyoto Encyclopedia of Genes and Genomes (KEGG) and Universal Protein (Uniprot) database, was used to perform GO and KEGG pathway enrichment analysis of DEGs. GO analysis involved three categories: biological process (BP), cellular component (CC) and molecular function (MF). P values <  0.05 was set as the thresholds for significant enrichment analyses.

Protein–protein interaction (PPI) network analysis

The Search Tool for the Retrieval of Interacting Genes/Proteins (STRING) database, a freely web-based analytic tool, can predict the interactions among proteins (Szklarczyk et al., 2019). Here, a PPI analysis was conducted to examine the interactive associations between protein products of DEGs. The Cytoscape software (http://www.cytoscape.org) was utilized to establish a PPI network. In addition, the CytoNCA (http://apps.cytoscape.org/apps/cytonca) was used to analyze topological characteristics of PPI network. The top 15 nodes were considered as hub genes according to the degree value.

TF-candidate gene network analysis

TFs can bind to specific DNA sequences in promoter region of target gene to regulate gene expression. The top 20 up- and down-regulated genes were regarded as the candidate genes. The DNA sequences (2 kb) in the upstream promoter region of these candidate genes were firstly downloaded from the University of California, Santa Cruz (http://www.genome.ucsc.edu/) databases. Then, we employed online match tool from TRANSFAC (http://genexplain.com/transfac) to predict potential TFs that targeted candidate genes. Notably, the TFs that had only one binding site with target genes were retained in this study. Finally, the Cytoscape software was used to build a transcriptional regulatory network and perform node degree analysis.

Evaluation the expression level and diagnostic values of key genes

Two published OS datasets were obtained from GEO database for the expression level evaluation of seven key DEGs (CAMP, METTL7A, TCN1, LTF, CXCL12, CYP4F3 and NETO2). Then, we performed a ROC analysis using the pROC package in R software (http://web.expasy.org/pROC/) to evaluate the diagnostic value of these seven DEGs. Accordingly, the area under the curve (AUC) was computed and the ROC curve was built. The AUC value > 0.8 showed a good diagnostic value for OS.

Results

Identification of DEGs

After data pre-processing, a total of 1,059 DEGs (569 up-regulated genes and 490 down-regulated genes) were identified between OS samples and normal controls according to methods described above. The clustering analysis indicated that top 100 DEGs could distinguish OS samples and controls from four datasets (Fig. S1). The top 20 up- and down-regulated genes were listed in Table 1.

Table 1 The list of top 20 up-regulated and down-regulated differentially expressed genes.

Gene symbol	Combined.ES	P_value	FDR	Up/Down-regulation	
NETO2	1.773	5.48E−10	1.16E−07	Up-regulation	
RTKN	1.664	3.90E−09	6.57E−07	Up-regulation	
TMEM65	1.669	1.08E−08	1.54E−06	Up-regulation	
MLLT11	1.507	1.41E−08	1.84E−06	Up-regulation	
LAPTM4B	1.499	1.78E−08	2.21E−06	Up-regulation	
ZC3H8	1.563	1.97E−08	2.41E−06	Up-regulation	
SLC35F2	1.570	3.46E−08	3.90E−06	Up-regulation	
IRX2	1.534	5.03E−08	5.19E−06	Up-regulation	
ZNF593	1.404	7.11E−08	6.78E−06	Up-regulation	
FLAD1	1.451	7.54E−08	7.10E−06	Up-regulation	
KCNG1	1.456	1.32E−07	1.16E−05	Up-regulation	
FGD1	1.304	2.59E−07	2.04E−05	Up-regulation	
RPAP2	1.480	2.63E−07	2.06E−05	Up-regulation	
DYRK4	1.405	2.83E−07	2.20E−05	Up-regulation	
EDARADD	1.360	3.06E−07	2.33E−05	Up-regulation	
PDCD5	1.285	3.77E−07	2.80E−05	Up-regulation	
TMEM97	1.349	4.04E−07	2.95E−05	Up-regulation	
GNL2	1.311	4.14E−07	3.01E−05	Up-regulation	
HOXB6	1.527	4.47E−07	3.17E−05	Up-regulation	
ZZZ3	1.504	5.79E−07	3.99E−05	Up-regulation	
CAMP	−4.158	0	0	Down-regulation	
AHSP	−3.156	0	0	Down-regulation	
OLFM4	−3.108	0	0	Down-regulation	
LTF	−3.006	0	0	Down-regulation	
ADH1C	−2.814	0	0	Down-regulation	
CXCL12	−2.746	0	0	Down-regulation	
BPI	−2.556	0	0	Down-regulation	
HBD	−2.615	8.88E−16	5.32E−13	Down-regulation	
TCN1	−2.522	1.11E−15	6.40E−13	Down-regulation	
FABP4	−2.296	7.77E−15	4.31E−12	Down-regulation	
RAB37	−2.279	2.02E−14	1.08E−11	Down-regulation	
FCN1	−2.320	2.95E−14	1.53E−11	Down-regulation	
TMEM154	−2.279	3.13E−14	1.55E−11	Down-regulation	
METTL7A	−2.181	5.44E−14	2.47E−11	Down-regulation	
CYP4F3	−2.304	6.66E−14	2.94E−11	Down-regulation	
CAT	−2.109	3.44E−13	1.43E−10	Down-regulation	
CHL1	−1.974	5.51E−13	2.17E−10	Down-regulation	
TMEM132C	−1.860	5.19E−12	1.85E−09	Down-regulation	
SERPINB2	−3.082	5.64E−12	1.97E−09	Down-regulation	
SLC28A3	−2.118	8.26E−12	2.69E−09	Down-regulation	

GO and KEGG enrichment analysis of DEGs

The GO enrichment analysis of DEGs showed that a total of 214 GO terms were enriched, including 194 GO-BP terms, 15 GO-CC terms and 5 GO-MF terms (Table S1). Specifically, for GO-BP analysis, these DEGs were strongly associated with positive regulation of cell death, myeloid cell activation involved in immune response and regulation of protein kinase activity. Meanwhile, protein domain specific binding and transcription factor binding were significantly enriched GO-MF terms. Many DEGs were primarily involved in multiple GO-CC terms, such as anchored component of membrane and tertiary granule. The top 20 clusters of significantly enriched GO terms were displayed in Fig. 1. In addition, these DEGs were markedly enriched in 20 KEGG pathways such as regulation of lipolysis in adipocytes, protein processing in endoplasmic reticulum and pathways in cancer (Table 2). Notably, the top 20 up-regulated genes did not enrich in any KEGG pathway. However, four of top 20 down-regulated genes played vital roles in multiple significantly enriched KEGG pathways, including FABP4 (fatty acid binding protein 4), CXCL12 (C-X-C motif chemokine ligand 12), CXCL12 (C-X-C motif chemokine ligand 12) and CAT (Catalase). More specifically, FABP4 was involved in regulation of lipolysis in adipocytes and CXCL12 participated in pathways in cancer and axon guidance (Table 2). CYP4F3 was closely correlated with arachidonic acid metabolism pathway and CAT was significantly enriched in biosynthesis of amino acids and AMPK signaling pathway (Table 2).

Figure 1 Top 20 significantly enriched Gene Ontology terms of differentially expressed genes.

Table 2 The top 20 significantly enriched KEGG pathways.

ID	Term	P_value	Count	Gene Symbols	
hsa04923	Regulation of lipolysis in adipocytes	0.0001	10	ADRB2, FABP4, PDE3B, PIK3CD, PLIN1, PTGER3, IRS2, ABHD5, PNPLA2, ADCY4	
hsa04141	Protein processing in endoplasmic reticulum	0.0001	19	BAG1, EIF2S1, HSPA2, LMAN1, MAN1A1, MAP3K5, P4HB, EIF2AK2, RPN2, RRBP1, SSR2, PREB, SEC24A, SEC61G, UBQLN2, DNAJB11, UBQLN4, DNAJC1, UBE2J2	
hsa05200	Pathways in cancer	0.0002	34	BCL2L1, CASP8, CDKN2A, CEBPA, CKS1B, COL4A1, DVL1, E2F1, EPAS1, FGF7, FGF13, FLT3, MTOR, GLI2, GLI3, GNAQ, PIK3CD, PTEN, PTGER2, PTGER3, RARB, RXRA, CXCL12, SKP2, STAT5A, TGFB3, TGFBR2, VEGFA, NCOA4, FGF16, PIAS2, ARHGEF11, LEF1, ADCY4, SHC1, CALM1, CHEK1, ELK1, FDPS, GPS2, MSX2, RANBP1, TLN1, VCAM1, TLN2	
hsa01230	Biosynthesis of amino acids	0.0004	11	CTH, ENO1, GAPDH, PC, SHMT1, TALDO1, TKT, TPI1, SDS, RPIA, AADAT, CAT, MDH2, ME1, MCEE	
hsa04974	Protein digestion and absorption	0.0019	11	COL2A1, COL4A1, COL5A2, COL10A1, COL11A1, COL17A1, CPA3, SLC7A8, COL18A1, COL27A1, SLC16A10	
hsa04722	Neurotrophin signaling pathway	0.0022	13	CALM1, MAPK14, MAP3K5, NFKBIE, NTF3, NTRK2, PIK3CD, PRKCD, RPS6KA2, SHC1, MAGED1, PRDM4, IRAK3	
hsa04152	AMPK signaling pathway	0.0023	13	CPT1A, ELAVL1, MTOR, LEP, PIK3CD, PPP2R5A, PPP2R5D, PRKAB2, IRS2, CREB5, CAMKK2, STRADB, CREB3L1, PRKCD, PRKCE, PTEN, RPS6KA2, TBC1D4, JAK2, NFKBIE, RXRA, SOCS2, CAT, ADCY4, CALM1, ELK1, FLOT2, PDE3B, PRKAR2B, SHC1, INPP5K, HSPA2	
M00007	Pentose phosphate pathway, non-oxidative phase, fructose 6P =>ribose 5P	0.0026	3	TALDO1, TKT, RPIA	
hsa03060	Protein export	0.0028	5	OXA1L, SRP54, SRP72, SEC61G, SRPRB	
hsa04933	AGE-RAGE signaling pathway in diabetic complications	0.0041	11	COL4A1, MAPK14, JAK2, PIK3CD, PRKCD, PRKCE, STAT5A, TGFB3, TGFBR2, VCAM1, VEGFA, CD247, MTOR, NFKBIE, RXRA, STAT6, IL27RA, LHB, SHC1, SOCS2, AOX1, BCL2L1, IL5RA, LEP, PIAS2	
hsa04360	Axon guidance	0.0045	16	EFNA1, EFNA3, EFNA5, EFNB1, EPHA3, EPHB2, EPHB6, FES, PIK3CD, CXCL12, SEMA7A, SEMA3A, RHOD, SEMA4C, NTNG2, PLXNA4	
hsa04015	Rap1 signaling pathway	0.0053	18	CALM1, MAPK14, EFNA1, EFNA3, EFNA5, FGF7, FGF13, GNAQ, ITGB3, PFN2, PIK3CD, TLN1, VEGFA, FGF16, RAPGEF3, APBB1IP, TLN2, ADCY4, BCL2L1, COL2A1, COL4A1, MTOR, ITGA7, JAK2, PPP2R5A, PPP2R5D, PTEN, RXRA, TLR4, CREB5, CREB3L1, THEM4	
hsa00590	Arachidonic acid metabolism	0.0055	8	ALOX15, GGT5, GPX3, GPX7, LTA4H, CYP4F3, PLA2G5, PTGES, GSTA4, GSTM3, SRM, GGCT	
hsa04932	Non-alcoholic fatty liver disease (NAFLD)	0.0060	14	CASP8, CEBPA, EIF2S1, LEP, MAP3K5, NDUFB9, NDUFS5, PIK3CD, PRKAB2, RXRA, UQCRH, IRS2, NDUFB11, NDUFA4L2	
hsa00620	Pyruvate metabolism	0.0066	6	GLO1, HAGH, MDH2, ME1, PC, LDHD	
hsa04924	Renin secretion	0.0073	8	ADRB2, CALM1, GNAQ, PDE1A, PDE1C, PDE3B, PTGER2, CLCA4	
hsa04750	Inflammatory mediator regulation of TRP channels	0.0101	10	CALM1, MAPK14, GNAQ, PIK3CD, PRKCD, PRKCE, PTGER2, TRPA1, TRPV4, ADCY4, ADRB2, ATP2B4, PPP2R5A, PPP2R5D, CREB5, RAPGEF3, CACNG7, CREB3L1, HSPA2, SHC1, PLA2G5, PTGIR, ARHGEF11, PPP1R14A, MYL6B, GPX3, GPX7, SLC26A4, ELK1, GNRH2, LHB	
hsa01524	Platinum drug resistance	0.0144	8	BCL2L1, CASP8, CDKN2A, GSTA4, GSTM3, MAP3K5, PIK3CD, PMAIP1, BCL2A1, DFFA, DFFB, EIF2S1, PARP2, DIABLO, DAB2IP, MAPK14, VCAM1, CREB5, CREB3L1, MLKL, CALM1, ITGB3, PLAT, VEGFA, TRPV4	
hsa04071	Sphingolipid signaling pathway	0.0146	11	MAPK14, MS4A2, GNAQ, MAP3K5, PIK3CD, PPP2R5A, PPP2R5D, PRKCE, PTEN, SPTLC2, SGMS1	
hsa00983	Drug metabolism-other enzymes	0.0147	6	CDA, CES1, DPYD, UCK2, CES2, UCKL1	
Notes.

KEGG; Kyoto Encyclopedia of Genes and Genomes.

PPI network analysis

To determine the relationships among DEGs, a PPI network was built based on the STRING database, which included 109 nodes and 196 protein pairs (Fig. 2). The top 15 hub genes contain PPBP (pro-platelet basic protein; degree = 13), CAMP (cathelicidin antimicrobial peptide, degree = 13), LTF (lactotransferrin, degree = 12), BST1 (bone marrow stromal cell antigen 1, degree = 12), CXCR2 (C-X-C motif chemokine receptor 2, degree = 10), OLFM4 (olfactomedin 4, degree = 10), STOM (stomatin, degree = 10), TCN1 (transcobalamin 1, degree = 9), SLC4A1 (solute carrier family 4 member 1, degree = 9), LTA4H (leukotriene A4 hydrolase, degree = 9), S100A9 (S100 calcium binding protein A9, degree = 9), CXCL12 (degree = 8), CLEC12A (C-type lectin domain family 12 member A, degree = 8), RAB37 (RAB37, member RAS oncogene family, degree = 8) and METTL7A (methyltransferase like 7A, degree = 8). More notably, these genes were all down-regulated.

TF-target network analysis

TF exerts crucial roles in regulating the expression of target gene. Herein, we employed TRANSFAC to predict TFs that regulated 20 up-regulated and down-regulated DEGs. As shown in Fig. 3, the TF-target gene regulatory network contained 95 nodes (55 TFs and 40 genes) and 275 TF-gene pairs. The top 15 genes in TF-target network were CHL1 (cell adhesion molecule L1 like, down-regulation, degree = 16), SERPINB2 (serpin family B member 2, down-regulation, degree = 15), SLC28A3 (solute carrier family 28 member 3, down-regulation, degree = 11), ZC3H8 (zinc finger CCCH-type containing 8, up-regulation, degree = 10), LAPTM4B (lysosomal protein transmembrane 4 beta, up-regulation, degree = 10), DYRK4 (dual specificity tyrosine phosphorylation regulated kinase 4, up-regulation, degree = 9), METTL7A (solute carrier family 28 member 3, down-regulation, degree = 9), KCNG1 (potassium voltage-gated channel modifier subfamily G member 1, down-regulation, degree = 9), CYP4F3 (down-regulation, degree = 9), GNL2 (G protein nucleolar 2, up-regulation, degree = 8), HBD (hemoglobin subunit delta, down-regulation, degree = 8), TCN1 (down-regulation, degree = 8), ZZZ3 (zinc finger ZZ-type containing 3, up-regulation, degree = 8), NETO2 (neuropilin and tolloid like 2, up-regulation, degree = 8), and LTF (lactotransferrin, down-regulation, degree = 8). In addition, the top six TFs that covered the most downstream genes were exhibited in Table 3, which contained Pax-4, 1-Oct, Nkx2-5, HNF-4, FOXD3 and Pax-6. Interestingly, TCN1, CYP4F3, NETO2 and CXCL12 were regulated by Pax-6 (Table 3).

Evaluation the expression level and diagnostic values of key genes

An external dataset (GSE39262) was obtained from GEO database, which contained 10 human osteosarcoma cell lines and five untransformed cell lines samples. The platform for this dataset was GPL96 [HG-U133A] Affymetrix Human Genome U133A Array. GSE126209 was downloaded from the GEO database, which inclued 12 osteosarcomas tumors and 11 adjacent normal tissues samples. The platform for this dataset was GPL20301 Illumina HiSeq 4000. Seven key genes (CAMP, METTL7A, TCN1, LTF, CXCL12, CYP4F3 and NETO2) were selected to verify in GSE39262. Among them, CAMP, METTL7A, TCN1, LTF and CXCL12 acted as hub genes in PPI network. METTL7A, CYP4F3, TCN1, LTF and NETO2 were key genes in TF-gene network. Moreover, Pax-6 regulated four key genes (TCN1, CYP4F3, NETO2 and CXCL12). The gene differential expression analysis of the GSE39262 dataset revealed that NETO2 was significantly up-regulated while CXCL12, METTL7A and TCN1 were significantly down-regulated, which were consistent with our integration analysis (Fig. 4; Table S2). The gene differential expression analysis of the GSE126209 dataset displayed that NETO2 was significantly up-regulated while CXCL12 was significantly down-regulated, which were consistent with our integration analysis (Fig. 5; Table S3). ROC analysis is a commonly used method to evaluate the value of genetic diagnosis and has been used in previously biomedical works (Le et al., 2019; Le, Yapp & Yeh, 2019; Thi, Trang & Khanh, 2020). Additionally, the results of the GSE39262 dataset showed that five genes had good diagnostic values for OS (CXCL12, CYP4F3, METTL7A, NETO2 and TCN1; Fig. 6). The AUC of CXCL12 was 1.000 and the specificity and sensitivity of this model were 100.0% and 100%, respectively. The AUC of CYP4F3 was 0.840 and the specificity and sensitivity of this model were 80.0% and 80.0%, respectively. The AUC of METTL7A was 0.900 and the specificity and sensitivity of this model were 100.0% and 70.0%, respectively. The AUC of NETO2 was 0.860 and the specificity and sensitivity of this model were 100.0% and 80.0%, respectively. The AUC of TCN1 was 0.820 and the specificity and sensitivity of this model were 80.0% and 90.0%, respectively. Ultimately, the results of the GSE126209 dataset showed that four genes had good diagnostic values for OS (CXCL12, METTL7A, NETO2 and TCN1; Fig. 7). CXCL12 was 1.000 and the specificity and sensitivity of this model were 100.0% and 100%, respectively. The AUC of METTL7A was 0.856 and the specificity and sensitivity of this model were 100.0% and 83.3%, respectively. The AUC of NETO2 was 0.833 and the specificity and sensitivity of this model were 100.0% and 75.0%, respectively. The AUC of TCN1 was 0.879 and the specificity and sensitivity of this model were 81.8% and 83.3%, respectively.

Figure 2 Protein–protein interaction networks of differentially expressed genes.

Red and green ellipses represent up-regulated and down-regulated genes, respectively. The black borders indicate top 20 up-regulated and down-regulated genes.

Figure 3 Transcription factor-top 20 up-regulated and down-regulated genes network.

Diamonds and ellipses represent transcription factors and top 20 up-regulated and down-regulated genes, respectively. Red and green ellipses represent up-regulated and down-regulated genes, respectively.

Table 3 The top 6 TF that has the most downstream genes.

TF	Number∗	Gene Symbol	
Pax-4	31	FLAD1, ZZZ3, GNL2, EDARADD, RTKN, LTF, CAMP, CHL1, ADH1C, TMEM154, IRX2, FABP4, LAPTM4B, SLC28A3, FGD1, CXCL12, HBD, TCN1, CAT, SLC35F2, DYRK4, METTL7A, OLFM4, AHSP, RAB37, TMEM97, SERPINB2, CYP4F3, PDCD5, BPI, KCNG1	
1-Oct	22	FLAD1, GNL2, RPAP2, EDARADD, ZC3H8, CHL1, CAMP, IRX2, FABP4, TMEM65, SLC28A3, HBD, TCN1, CAT, TMEM132C, DYRK4, METTL7A, TMEM97, HOXB6, SERPINB2, BPI, KCNG1	
Nkx2-5	19	FLAD1, MLLT11, RPAP2, EDARADD, CAMP, CHL1, TMEM154, LAPTM4B, TMEM65, SLC28A3, FGD1, TCN1, SLC35F2, DYRK4, AHSP, SERPINB2, CYP4F3, KCNG1, BPI	
HNF-4	16	ZZZ3, ZNF593, GNL2, LTF, CHL1, CAMP, LAPTM4B, SLC28A3, FGD1, CXCL12, HBD, SLC35F2, METTL7A, RAB37, CYP4F3, KCNG1	
FOXD3	14	GNL2, RPAP2, EDARADD, ZC3H8, CHL1, LTF, FABP4, FGD1, HBD, DYRK4, METTL7A, OLFM4, NETO2, RAB37	
Pax-6	13	EDARADD, GNL2, CHL1, TMEM65, SLC28A3, CXCL12, HBD, TCN1, DYRK4, NETO2, SERPINB2, CYP4F3, KCNG1	
Notes.

Number∗ indicates the number of genes regulated by the TF

TF transcription factor

Figure 4 Box plots of seven differentially expressed genes in the GSE39262 dataset.

The x-axes represent control and case groups while the y-axes represent the relative expression levels of the genes. Seven genes included NETO2, CAMP, METTL7A, TCN1, LTF, CXCL12 and CYP4F3.

Figure 5 (A-G) Box plots of seven differentially expressed genes in GSE126209 dataset.

The x-axes represent control and case groups while the y-axes represent the relative expression levels of the genes. Seven genes included NETO2, CAMP, METTL7A, TCN1, LTF, CXCL12 and CYP4F3.

Figure 6 ROC curves of selected differentially expressed genes in the GSE39262 dataset.

The x-axes and the y-axes show 1-specificity and sensitivity, respectively. ROC, receiver operating characteristic. (A-G) The seven genes included NETO2, CAMP, METTL7A, TCN1, LTF, CXCL12 and CYP4F3.

Figure 7 ROC curves of selected differentially expressed genes in the GSE126209 dataset.

The x-axes and the y-axes show 1-specificity and sensitivity, respectively. ROC, receiver operating characteristic. (A-G) The seven genes included NETO2, CAMP, METTL7A, TCN1, LTF, CXCL12 and CYP4F3.

Discussion

OS is a common malignant bone tumor and originates from mesenchymal stromal cells (MSCs) (Xiao, Hogendoorn & Cleton-Jansen, 2013). The heterogeneous histopathological characteristics and complex genomic landscape of OS have been major challenges for elaborating underlying the molecular pathogenesis of OS. In this study, we included four OS datasets and identified 1,059 DEGs (569 up-regulated DEGs and 490 down-regulated DEGs) between OS and normal samples. These genes were significantly enriched in 54 KEGG pathways such as pathways in cancer. Moreover, CAMP, METTL7A, TCN1, LTF and CXCL12 served as hub genes in PPI network. METTL7A, CYP4F3, TCN1, LTF and NETO2 were key players in TF-target gene regulatory network. Interestingly, TCN1, CYP4F3, NETO2 and CXCL12 were all regulated by Pax-6. Additionally, the expression patterns of key genes (CAMP, METTL7A, TCN1, LTF, CXCL12, CYP4F3 and NETO2) were selected to verify in two published OS datasets (GSE39262 and GSE126209).

CAMP, also known as hCAP18 or LL37, is an antimicrobial peptide gene in human (Larrick et al., 1995). The C-terminal of the protein product of CAMP contains a 37-amino acid-long peptide with broad spectrum-antibacterial activity (Vandamme, Luyten & Schoofs, 2012). There are positive expressions of CAMP in the multiple cell systems, such as epithelial cells, neutrophils and macrophages (Dhawan et al., 2015; Frew et al., 2014; Li et al., 2018). Wu et al. (2012) suggested that bone marrow stroma could express CAMP, which may be a potential ex vivo priming factor for hematopoietic stem progenitor cells to promote hematopoietic reconstitution after transplantation. Later, Coffelt et al. (2019) discovered that CAMP expression level was elevated in MSCs compared to that in ovarian cancer cells. Herein, our analysis showed that CAMP was the most down-regulated gene in patients suffering from OS. Besides, CAMP acted as a hub gene in PPI network, suggesting that this gene may be involved in the pathologic mechanism of OS. Although the underlying role of CAMP on the initiation and progression of OS has not been investigated, available evidence showes that CAMP plays significant roles in several cancers, including breast cancer, lung cancer and pancreatic cancer (García-Quiroz et al., 2016; Sainz Jr et al., 2015; Von Haussen et al., 2008). More notably, existing data indicated that CAMP had either carcinogenic or anti-cancer effects (Chen et al., 2018; Wu et al., 2010). Therefore, the influence of CAMP on OS occurrence and development needs to be further clarified in future.

Our gene differential expression revealed that CXCL12 and TCN1 were down-regulated in OS patients, which were verified in a validation dataset. Moreover, these two genes also acted as hub genes in PPI network. In addition, up-regulated NETO2 and down-regulated CYP4F3 had high degree in TF-gene regulatory network. Interestingly, CXCL12, TCN1, NETO2 and CYP4F3, regulated by Pax-6, exhibited important diagnostic values for OS. CXCL12 is also called stromal cell-derived factor-1 (SDF-1) and can bind to G-protein-coupled chemokine receptor CXCR4 (Nagasawa, 2014). Increasing studies suggested that CXCL12/CXCR4 axis played pivotal roles in tumor growth and development (Balkwill, 2004; Lu et al., 2015; Perissinotto et al., 2005). Li et al. (2018). highlighted that epigenetic regulation of CXCL12 by DNA methyltransferase 1 was associated with the metastasis and immune response in OS. Previous reports also indicated that down-regulation of CXCR4 induced OS cell apoptosis via suppressing PI3K/Akt/NF-κβ pathway (Pollino et al., 2019). However, there is no directive evidence to support the involvement of TCN1, NETO2 and CYP4F3 in OS. Notably, Pax-6 is a highly conserved evolutionarily TF and belongs to paired box TF family (Mansouri & Gruss, 1996). Several studies have pointed out that Pax6 participated in the regulation of cancer cell proliferation and progression (Shyr et al., 2010; Zong et al., 2011). Yang et al. (2019) established a TF-top 20 DEGs regulatory network by integrating and analyzing three GEO datasets (GSE66673, GSE49003 and GSE37552), and found that Pax-6 down-regulated BMP6 expression in non-metastatic OS samples. Taken together, we inferred that CXCL12/TCN1/NETO2/CYP4F3-Pax-6 axis may be implicated in the pathogenesis of OS, and four genes (CXCL12, TCN1, NETO2 and CYP4F3) were novel diagnostic biomakers for OS.

METTL7A and LTF are reported to act as tumor suppressor genes (Qi et al., 2017; Zhang et al., 2011; Zhang et al., 2015). Similarly, our findings showed that the expressions of METTL7A and LTF were decreased in OS samples. Moreover, these two genes were both hub genes in PPI analysis and key gene nodes in TF-gene regulatory analysis. These results implied that METTL7A and LTF may be correlated with underlying mechanisms of OS. However, the potential effects of METTL7A and LTF down-regulation on OS progression needs to be further investigated.

Although we have identified multiple novel gene signatures associated with OS, there are still limitations in this work. Our conclusion was drawn based on an integrated bioinformatic analysis. Therefore, additional experiments are required to confirm our findings. In addition, a larger sample size verification will also improve the reliability of our conclusion. Moreover, the clinical information should be collected to evaluate the diagnostic value of biomarkers for OS patients. Finally, the biological significances of key biomarkers will be investigated in model systems or cell lines.

In summary, a total of 1,059 DEGs were identified between OS and normal samples. Among them, up-regulation of NETO2 and down-regulation of METTL7A, TCN1, and CXCL12 may be potential gene signatures related to OS. Pax-6 was also probably associated with the pathological process of OS. However, a comprehensive bioinformatics analysis with larger sample size and in vivo or in vitro assays should be performed to confirm our results.

Supplemental Information

Supplemental Information 1 The heat map of the top 100 differentially expressed genes

Click here for additional data file.

Supplemental Information 2 The information from selected GEO datasets in this study

Click here for additional data file.

Supplemental Information 3 The differential expression of seven genes in GSE39262 dataset

Click here for additional data file.

Supplemental Information 4 The differential expression of seven genes in GSE126209 dataset

Click here for additional data file.

Additional Information and Declarations

Competing Interests

Author Contributions

Data Availability

The authors declare there are no competing interests.

Yutao Jia conceived and designed the experiments, performed the experiments, analyzed the data, prepared figures and/or tables, authored or reviewed drafts of the paper, and approved the final draft.

Yang Liu performed the experiments, analyzed the data, authored or reviewed drafts of the paper, and approved the final draft.

Zhihua Han performed the experiments, analyzed the data, prepared figures and/or tables, authored or reviewed drafts of the paper, and approved the final draft.

Rong Tian conceived and designed the experiments, analyzed the data, authored or reviewed drafts of the paper, and approved the final draft.

The following information was supplied regarding data availability:

The data is available at NCBI GEO: GSE12865, GSE19276, GSE87624, GSE99671, GSE39262, and GSE126209.

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
