# Peer review of "Identification of potential gene signatures associated with osteosarcoma by integrated bioinformatics analysis"

_PeerJ, doi:10.7717/peerj.11496_

## Round 0.1 · original submission · Major Revisions

Several significant areas of concern regarding the analyses need to be addressed if you decide to revise and resubmit your manuscript. First, please correct for batch effects when analyzing disparate data sets from GEO. Second, it is not clear why you restricted your analysis to microarray data given the significant number of relevant RNA-seq data sets available. Extending your analysis to include these data sets would provide a significant increase in power, and they should be included. Finally, please clarify any biological insights gained from your work that may complement or expand upon previous studies.

Reviewer 1 ·

Basic reporting

NA

Experimental design

NA

Validity of the findings

NA

Additional comments

In this manuscript “Identification of potential gene signatures associated 1 with osteosarcoma by integrated bioinformatics analysis” Jia et al. have identified the differentially expressed genes in osteosarcoma patients and performed functional enrichment analysis to identify certain genes with prognostic value. Authors have performed multiple bioinformatics analysis but biological insights gained from the analysis is not discussed. Some of the major comments that need to be addressed are:

1. It is not clear why autjors have relied on microarray data when TCGA has RNA-seq information of almost 100 osteosarcoma patients. Reanalysis with RNA-seq data will show the robustness of the results.

2. Authors have used different datasets from GEO. How these different datasets were batch corrected and normalized together and differentially expressed genes were identified?

3. What does color represent in Figure 1. Font of the text in the figure is very small and unreadable. What does these network diagrams in panel B represent? How these top genes cluster together? A heatmap representation of top DEGs could help in understanding the variations in the expression levels of these genes acorss various patients.

4. Are these identified DEGs enriched to be hub genes in protein interaction map analysis? STRINg includes many other interactions than just experimentally validated interaction data. Similar analysis across various other datasets (e.g. IrefWeb, HumanNet) could give a clear picture of the interaction patterns of these DEGs.

5. Figure4: Indicate the p values of significance. Also, show the number of samples in each boxplot. Instead of relative expression level, absolute expression level can clearly show if the differences in expression are robust in an independent dataset.

6. ROC analysis is not informative as numbers of samples are very low. Does any of these genes are predictive of survival status in osteosarcoma patients? Are these genes useful to predict drug response in these patients?

7. There are spelling mistakes at some places in the manuscript and English needs some correction.

Reviewer 2 ·

Basic reporting

1. Relevant prior literature has not been appropriately referenced. For example,
Identification of potential crucial genes and key pathways in osteosarcoma PMID: 32665038
This study is very similar with the paper I mentioned above.
2. In Figure4, why the authors show box plots of seven differentially expressed genes, why not other DEGs? and no statistical analysis was used to compare, this data is of no meaning.
3. The quality of Figure 5 is poor.

Experimental design

This work aims to 20 screen novel potential gene signatures associated with OS using bioinformatics anaysis.
1. The analysis and design is very traditional and lack of novelty.
2. The authors downloaded four datasets from the GEO database, however, the detailed information of these datasets has not been provided. Are these datasets from the same platform?
3. The authors should make it clear how did they processed the four datastes? If these datasets come from different platform? How did they combining these datasets? Has batch correction been down? I did't see the quality control data in this manuscript
4. What is the cutoff value of Fold change of DEGs were not mentioned .
5. What is the significance of hierarchical clustering analysis of top 100 DEGs?
6. What is the meaning of top15 nodes? The authors should make it clear how and why they choose top 15 , but not top 20? or top 10? top 5? they should also clearly define how the top 15 nodes were choose?
7. Why the top 20 up- and down-regulated genes were regarded as the candidate genes? Why not 10 up- and down? What is the reason.
8. In Validation of DEGs and ROC analysis section, the authors state they validate the DEGs using ROC, I do not agree this is validation. It is just another kind of show of the DEGs. It is not a validation.

Validity of the findings

1. Expression of microarray data is used to be correlated with bias, and the sample used in this study is very limited. If possible, the authors should use other independent datasets to validate their results.
2. One major issue is the batch effect between the four datasets.

Additional comments

1. In fact, this paper is very similar to published articles.
2. The authors should clearly point out what contributions they did ? The potential genes? this data is not very solid.

Reviewer 3 ·

Basic reporting

English language should be improved. There are some parts that were unclear and ambiguous. Also, there contain some grammatical errors and typos such as:
- High-throughout sequencing ...
- Sun et al evaluated difference ...
- ... biomakers for OS ...
- ... is a antimicrobial peptide gene ...
- ... which may be as a potential ...
- ... these two gene were both ...
- ...
Therefore, the authors should re-check and revise carefully. It is better to be checked by a native speaker or English editing service.

The authors should provide more background in the abstract and introduction. Now the motivation or objective of the study is unclear.

Quality of figures should be improved.

Experimental design

What is "porgn"?

The authors should explain clearly how they considered the inspection and removal of batch effects when merging five sets of data from GEO.

The authors used R to implement the analyses. It is important to provide the source codes to help reproducing the results.

ROC curve & AUC have been used in previously biomedical works such as PMID: 31277574, PMID: 31921391, and PMID: 32613242. Therefore, the authors should refer more works into this description to attract broader readership.

Validity of the findings

What is the cut-off value for GO enrichment analysis?

In the Fig. 5's legend, the authors mentioned x‑axes and the y‑axes show specificity and sensitivity, respectively. However I think x-axes should be "1-specificity".

The authors should compare the performance between 7-genes signature and all-genes signature in terms of ROC curve or AUC, or pathway analyses.

How about prognostic biomarkers for these genes?

The authors convinced some different genes from previous works on OS, then they should compare the performance results with the previous works and to see which gene set is really significant in this disease.

Additional comments

No comment

---

## Round 0.2 · Major Revisions

Thank you for your patience regarding your recent manuscript revisions. After careful review, there remains one significant issue with your submission that requires attention before being considered for publication. As earlier reviewers noted, the sample size in your study is very low; consequently, the use of ROC to estimate the diagnostic potential of genes of interest is quite problematic. It is important that you address this issue directly, preferably by including a power analysis and likely by including additional data sets. There are numerous approaches to guide you here, including (for example):

https://www.ncbi.nlm.nih.gov/pmc/articles/PMC5121784/

Reviewer 2 ·

Basic reporting

The same with previous review.

Experimental design

The same with previous review.

Validity of the findings

The same with previous review.

Additional comments

The authors failed to response to my review comment carefully.

All comments should be considerd carefully instead of making a perfunctory effort.

Reviewer 3 ·

Basic reporting

No comment

Experimental design

No comment

Validity of the findings

No comment

Additional comments

My previous comments have been addressed satisfactorily.

---

## Round 0.3 · Major Revisions

Please address the original comments from reviewer 2. Thank you!

---

## Round 0.4 · accepted · Accept

Thank you very much for your patience as we worked through the review process of your manuscript.